computational biology, evolution, microbiology

microbial evolution, trait correlations, trait adaptation, phytoplankton, biogeochemistry, principal component analyses

**Author for correspondence:**
Naomi M. Levine
e-mail: n.levine@usc.edu

# The evolution of trait correlations constrains phenotypic adaptation to high $CO_2$ in a eukaryotic alga

Nathan G. Walworth[1], Jana Hinners[2], Phoebe A. Argyle[3], Suzana G. Leles[1], Martina A. Doblin[3], Sinéad Collins[2] and Naomi M. Levine[1]

[1]Department of Biological Sciences, University of Southern California, Los Angeles, CA 90089-0371, USA
[2]Institute of Evolutionary Biology, University of Edinburgh, Edinburgh EH9 3FL, UK
[3]Climate Change Cluster, University of Technology Sydney, Sydney, New South Wales 2007, Australia

NGW, 0000-0002-7408-9461; SGL, 0000-0002-7819-5851; SC, 0000-0003-3856-4285; NML, 0000-0002-4963-0535

Microbes form the base of food webs and drive biogeochemical cycling. Predicting the effects of microbial evolution on global elemental cycles remains a significant challenge due to the sheer number of interacting environmental and trait combinations. Here, we present an approach for integrating multivariate trait data into a predictive model of trait evolution. We investigated the outcome of thousands of possible adaptive walks parameterized using empirical evolution data from the alga *Chlamydomonas* exposed to high $CO_2$. We found that the direction of historical bias (existing trait correlations) influenced both the rate of adaptation and the evolved phenotypes (trait combinations). Critically, we use fitness landscapes derived directly from empirical trait values to capture known evolutionary phenomena. This work demonstrates that ecological models need to represent both changes in traits and changes in the correlation between traits in order to accurately capture phytoplankton evolution and predict future shifts in elemental cycling.

## 1. Introduction

Microbes play a critical role in regulating biogeochemistry and the global climate. In recent years, there has been a significant increase in global change studies examining the role of microbial evolution in shaping future biogeochemical cycles. This work has helped to more explicitly integrate the fields of evolution and microbial ecology, resulting in both long-term experimental evolution studies with ecologically important microbes and, to a limited extent, the incorporation of adaptation into ecological and ocean circulation models [1–13]. These studies are just the first step in tackling the immensely complex challenge of microbial evolution and its influence on global biogeochemistry. We still have only a limited understanding of how microbial communities will respond to multi-stressor and fluctuating environmental change. Additionally, the sheer number of interacting environmental and trait combinations exceeds our experimental ability to robustly quantify these responses [14,15]. Hence, experimental and theoretical methods to reduce dimensionality and extract broad evolutionary patterns across traits and taxa are critical for creating a framework that can both help guide experiments and make more accurate future predictions [5].

Seminal research in quantitative genetics has investigated the impact of trait variation, genotypic variability, inheritance, epistasis and environmental variability on adaptive walks using multivariate and eigenvector methods on theoretical populations experiencing environmental change [16–20]. These studies have broadly found that an evolving population may be able to access only a subset of phenotypes. Other theoretical approaches emphasized the role of de novo mutations in a fitness landscape without accounting for standing genetic

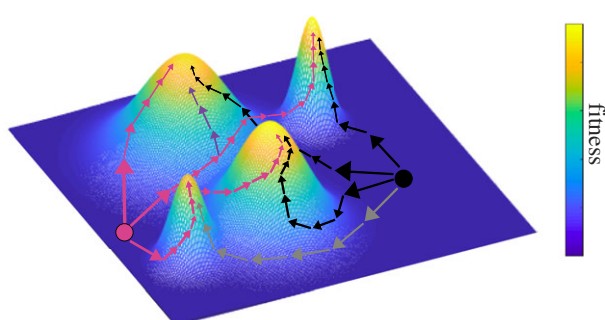

**Figure 1.** Comparison of adaptive walks for two different phenotypes in a rugged fitness landscape with four high-fitness peaks. Two example starting phenotypes are represented as circles (magenta and black). The *x*- and *y*-axis represent dimensions in fitness space (e.g. different traits). The phenotypes start with low fitness (*z*-axis) and through trait and trait correlation changes move to higher fitness. The adaptive walk is governed by historical bias, or different initial trait architecture, that impacts the movement of the population within the landscape. As the adaptive walk proceeds, the population moves to the top of one of the fitness peaks. While there are several paths available to each starting phenotype (represented by magenta and black arrows), due to historical bias (trait correlation constraints), some paths can be inaccessible (denoted by the grey and purple arrows). Note that depending on historical bias and the phenotypic starting location, some high-fitness peaks are either more difficult to access or completely inaccessible.

variation [21,22]. While these studies have generated evolutionary theory [22,23] relevant for marine microbes, the gulf between evolution literature and biological oceanography has widened such that current state-of-the-art evolutionary models are not suited to interface with biogeochemical ODE models [24].

Empirical data have also demonstrated that biological systems produce certain phenotypic variants more readily than others in response to a perturbation (mutation or environmental change) due to the inherent structure, composition and evolutionary history of a population [25,26]. These findings contrast with the long-held assumption of isotropic (i.e. equal) variation [27] and have revealed instead that only a limited part of multivariate phenotypic space (i.e. only certain phenotypes) can be accessed [18,20,23,25–28]. Figure 1 shows an illustrative example of how the accessibility of high-fitness phenotypes (i.e. peaks in a fitness landscape) differs depending on the starting location (ancestral phenotype) and the initial trajectory, which is dictated by a population's collective genetic architecture (figure 1, magenta and black circles and paths). In summary, a growing body of the literature has shown that genetic architecture influences how traits and trait correlations are impacted by environmental shifts and that these shifts produce non-random distributions of phenotypes [29–31]. These studies have substantial consequences for understanding future shifts in marine phytoplankton function and thus carbon cycling and global climate [32,33]. However, there have been few attempts to investigate the implications of this phenomenon for the evolution of trait and trait correlations of photosynthetic microbes [19,20].

We introduce a framework for understanding how evolutionary trajectories of phenotypes (suites of traits) in microbial populations are impacted by the evolutionary relationships between traits (historical bias) and the evolution of trait relationships. Here, we define bias as standing trait correlations (i.e. relationships) in a population that are heritable

and can impact fitness such that, over time, these correlations can constrain the direction of evolution [16]. Critically, this framework is based on measurable trait values and so does not require the quantification of the fitness impacts of mutations. This is important as most marine microbes are not genetically accessible and mapping the fitness effects of individual genetic variants is resource intensive. Our trait correlation evolution (TRACE) model is a first step towards investigating how correlated metabolic traits with clear biogeochemical significance may impact elemental cycling under environmental change (e.g. ocean acidification). Specifically, TRACE generates a trait-based fitness landscape based on empirical trait data thus facilitating future integration into global biogeochemical models in which marine microbes are represented using trait-based functional groups. Our results indicate that populations harbouring trait correlations oriented in (i.e. consistent with) the direction of selection may experience accelerated rates of adaptation. We suggest that incorporating these dynamics into biogeochemical models will be important for accurately predicting the impact of microbial adaptation on rates of biogeochemical cycling in the ocean.

## 2. Material and methods

### (a) Trait-based fitness landscape

We generated a multivariate trait-based fitness landscape (trait-scape) using principal component analyses (PCA). Specifically, a trait-scape was created using four independent ecologically relevant traits (growth rate, respiration, cell size and daughter cell production) from five genotypes of high-$CO_2$-adapted *Chlamydomonas reinhardtii* [3]. PCA was conducted on standardized evolved traits resulting in a total of 86% of variance explained on two axes, 54% and 32% on axes PC1 and PC2, respectively (figure 2*a*). To select a start and endpoint for the adaptive walk, ancestral populations were projected onto the evolved PC axes. A single genotype was selected for the modelling exercise where the observed ancestral trait values defined the start point of the adaptive walk (tan circle in figure 2*b*; electronic supplementary material, file S1) and the corresponding evolved population trait values defined the evolutionary endpoint (red circle in figure 2*b*; electronic supplementary material, file S1). Additional simulations were conducted in which the start point was varied. As the specific traits themselves were not relevant for this study, we will hereafter refer to them as traits 1–4. We refer the reader to [3] for an in-depth discussion of the evolution experiment.

### (b) Trait correlation evolution model dynamics

The TRACE model framework simulates the adaptive walk of a microbial population across a trait landscape (trait-scape) towards a high-fitness area. TRACE was adapted from an individual based Fisher model of adaptation [1,34,35]. Each generation, each individual in the population experienced either a change in traits or changes in traits and trait correlations. Changes in trait values moved these individuals across the trait-scape while trait correlations constrained the direction of movement. Selection was imposed based on distance to the evolutionary endpoint in the trait-scape (described below), such that the population evolved towards the high-fitness region of the trait-scape. In essence, this framework selected for individuals with the smallest overall difference across all trait values from the empirically observed high-fitness phenotype. The weighting of the traits was derived from the observed evolved phenotypes evaluated using PCA, such that traits that were not observed to play an

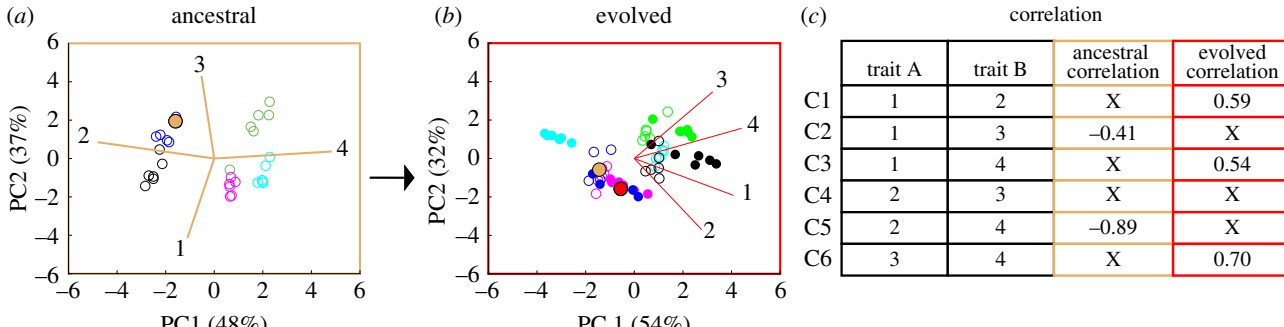

**Figure 2.** Principal component analysis (PCA) of ancestral and evolved trait values and the corresponding trait correlations. (*a*) Ancestral PCA calculated from the values of four ancestral traits across five genotypes where each point represents an independent biological population (i.e. culture) coloured by genotype. Percentages associated with PC1 and PC2 denote the amount of variance explained by each PC axis. (*b*) Evolved PCA plot calculated from the evolved values of the same four traits as in (*a*) across five genotypes. Filled circles represent the independent populations of the evolved genotypes. Open circles represent the corresponding populations of the ancestral genotypes (*a*) projected onto the evolved PC axes. The tan and red filled circles denote the start and end coordinate of the model, respectively. (*c*) Table of all six possible trait combinations and their values in their ancestral and evolved genotypes. An 'X' indicates a non-statistically significant trait correlation ($p > 0.05$).

important role in fitness in the high-$CO_2$ environment had low weight. It is important to note that the model did not directly select for trait correlations, but that specific correlations emerged in the population if they provided fitness advantage in terms of trait dynamics.

In the default model simulations (referred to as 90/10), in each generation 90% of individuals were randomly chosen to experience a random change in a trait value (while maintaining all existing trait correlations) while 10% experienced both a trait and trait correlation change. These changes were drawn from a Gaussian distribution (mean = 0 and standard deviation = 0.05) such that small changes were common and large changes were rare. For each individual, the randomly chosen trait change was added to the existing trait value. Following this first trait change, the remaining three trait values were updated using the trait correlations for that individual in that time step. For example, if trait 1 was initially changed, then traits 2, 3 and 4 would subsequently be updated by multiplying the new trait 1 value by the three trait correlations (1v2, 1v3 and 1v4).

The remaining 10% of the population experienced both a trait and a trait correlation change. For each individual, one of the six trait correlations was randomly selected to change. Similar to the trait change, a random value was drawn from a Gaussian distribution with a mean of 0 and standard deviation of 0.05 and added to the existing correlation value. Next, one of the two traits associated with that correlation was randomly chosen and a trait change was selected in the same manner as above. Finally, the second trait tied to the correlation was updated using the new correlation and trait value (the other modelled trait values were not updated in this generation). These trait changes moved the individuals within the trait-scape.

Selection was imposed using the Euclidean distance of each individual to the high-fitness area ($z_i$). Specifically, the fitness of each for each individual ($w_i$) was calculated as [1,34]

$$w_i = e^{-z_i^2/2}. \tag{2.1}$$

Individuals were then randomly sampled with replacement weighted by $w_i$ such that high-fitness individuals were more likely to persist to the next generation, as in [1,34].

## (c) Model simulations
The model trait-scape and the high-fitness area (red circle in figure 2*b*) were defined based on empirical data from the *Chlamydomonas* long-term evolution study [3]. The model was initialized with a population of 1000 individuals with either (i) all

individuals containing the same trait values corresponding to the empirically observed ancestral trait values (tan circle in figure 2*b*), (ii) all individuals containing the same trait values corresponding to an alternative starting location or (iii) a mixed population with multiple different starting trait values based on the empirical data. To explore the impact of historical bias, trait correlations in the starting populations were initialized in three different ways: mixed mode, ancestral mode and evolved mode (described below). The modes differed only in the initial conditions not in model dynamics. Each model run was conducted for 2000 generations with 100 replicates each. All model parameters are given in electronic supplementary material, table S1. Previous work by us and others have demonstrated that adaptive outcomes using this framework are robust across a wide range of population sizes (electronic supplementary material, information S1) [1,34]. Several sensitivity studies were conducted to test model dynamics (see electronic supplementary material, information S2).

### (i) Mixed mode (no bias) simulations
To test all possible adaptive walks between the ancestral start point and the evolutionary endpoint, simulations were conducted with no historical bias. Specifically, each individual was assigned correlation values randomly chosen from a standard uniform distribution over the interval (−1,1). Hence, every individual started with the same four trait values but random correlation values. See electronic supplementary material, information S3 for additional model detail.

### (ii) Ancestral mode simulations
To test the effects of systematically adding ancestral bias, four ancestral sub-modes were conducted: A1, A2, A3 and A4. For sub-mode A1, random correlation values were generated as above for five of the six trait correlations, and one empirical ancestral correlation was used for all individuals. For A2, A3 and A4, all steps were the same except that two, three and four empirical ancestral correlations were added back, respectively. For simplicity, we chose to sequentially add in ancestral correlations based on the empirically calculated significant trait correlations from most significant to least significant ($R^2 = -0.89$ to $0.54$; figure 2*c*).

### (iii) Evolved mode simulations
The same procedure described above for the ancestral mode was conducted for the evolved mode, but instead empirical evolved correlations were systematically added (modes E1–E4).

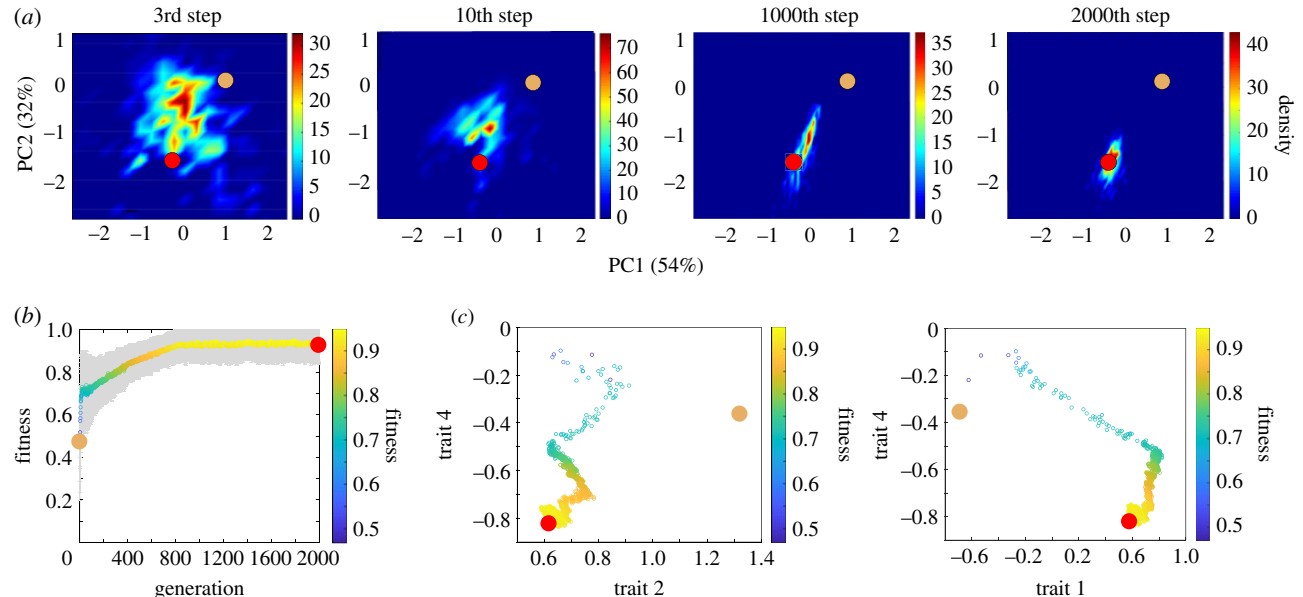

**Figure 3.** Representative adaptive walk in the evolved trait-scape of a population of 1000 individuals. (*a*) Density plots of an adaptive walk of a single population for a single run (*n* = 1000 individuals) starting at the tan dot and ending at the red dot. Each plot represents a different point in time (i.e. generation) in the adaptive walk with the colour representing the density of individuals in a given area. (*b*) Fitness plot of the population across the entire adaptive walk with the coloured line and grey region representing the mean and standard deviation, respectively. Both the *y*-axis and colour indicate fitness. (*c*) Trait versus trait plots representing the same adaptive walk where lower fitness denotes the start of the walk and higher fitness denotes the end. As in (*b*), each point represents the mean standardized trait value of all individuals at a specific generation, or step.

## (d) Hierarchical clustering

Hierarchical clustering with multiscale bootstrap resampling (1000 replicates) on mean trait correlation values was conducted with R package pvclust [36] using Euclidean distance and the average (UPGMA) method. Principal component analysis using mean correlation values was conducted with R package vegan [37], and pvclust clusters with approximately unbiased (AU) *p*-values < 75% were projected onto the PC coordinate plane as convex hulls.

## 3. Results

### (a) Multi-dimensional trait evolution

To understand the constraints on phenotype evolution, we must consider how multi-dimensional traits are altered by selection. Previous empirical work has shown that both trait values and the correlations between traits are altered as a population adapts to a new environment. For example, when five genotypes of the green alga *C. reinhardtii* were selected under high $CO_2$ [3], all quantified traits changed to varying degrees depending on the genotype and the correlations between many traits changed with some traits becoming correlated (e.g. 1v2) while others becoming uncorrelated (e.g. 2v4; figure 2*c*).

To understand how trait movement within the trait-scape can be constrained by historical bias (correlations between traits in the ancestral population), we developed a statistical model (TRACE) of multi-trait adaptation and investigated probabilities of different emergent evolutionary outcomes. We began with a 'null hypothesis' model in which there was no historical bias (mixed mode simulation) and then systematically added in bias to determine the impact on population level adaptation. An example of model dynamics from a single run in mixed mode is shown in figure 3*a* where a representative population consisting of a thousand individuals moved over time from the ancestral starting phenotype to the evolved high-fitness area (figure 3*a*). This resulted in an overall increase in fitness of the population over time (figure 3*b*). The underlying dynamics of the model (changes in trait values and trait correlation changes) for three representative traits are shown in figure 3*c*.

Though TRACE is a novel modelling framework for understanding trait adaptation, the model captures well known dynamics of adaptions. For example, fitness effects produced from changes at the beginning of the walk were significantly greater than at the end of the walk consistent with previous experimental and theoretical work [1,34,38–41] (figure 3). Although some individuals reached a maximum possible fitness of 1 (i.e. the evolutionary end coordinate), the mean population fitness consistently remained below 1 (figure 3*b*). This is due to the fact that the model is simultaneously optimizing multiple traits and their correlations, which inherently introduces small but significant amounts of persistent phenotypic variation. In addition, while the average movement of the population was fairly linear in PC space (figure 3*a*), the trajectory of trait changes was not linear (figure 3*c*).

### (b) Accessibility of cryptic phenotypes

Four distinct population types (i.e. traits + trait correlations for the final population) emerged across the replicate runs (1000 individuals × 100 replicate runs = 100 000 individuals total). We term these population types (Pop-MA, Pop-MB, Pop-MC and Pop-MD) 'cryptic phenotypes' as they had statistically similar end mean fitness values and occurred in the same region of the trait-scape but had distinct trait correlations and, to some extent, distinct trait values (figure 4; electronic supplementary material, figure S1). In other words, these cryptic phenotypes represent four distinct evolutionary outcomes of different trait correlations + trait values that all converged on the single evolutionary end coordinate in the evolved trait-scape. For some correlations such as trait

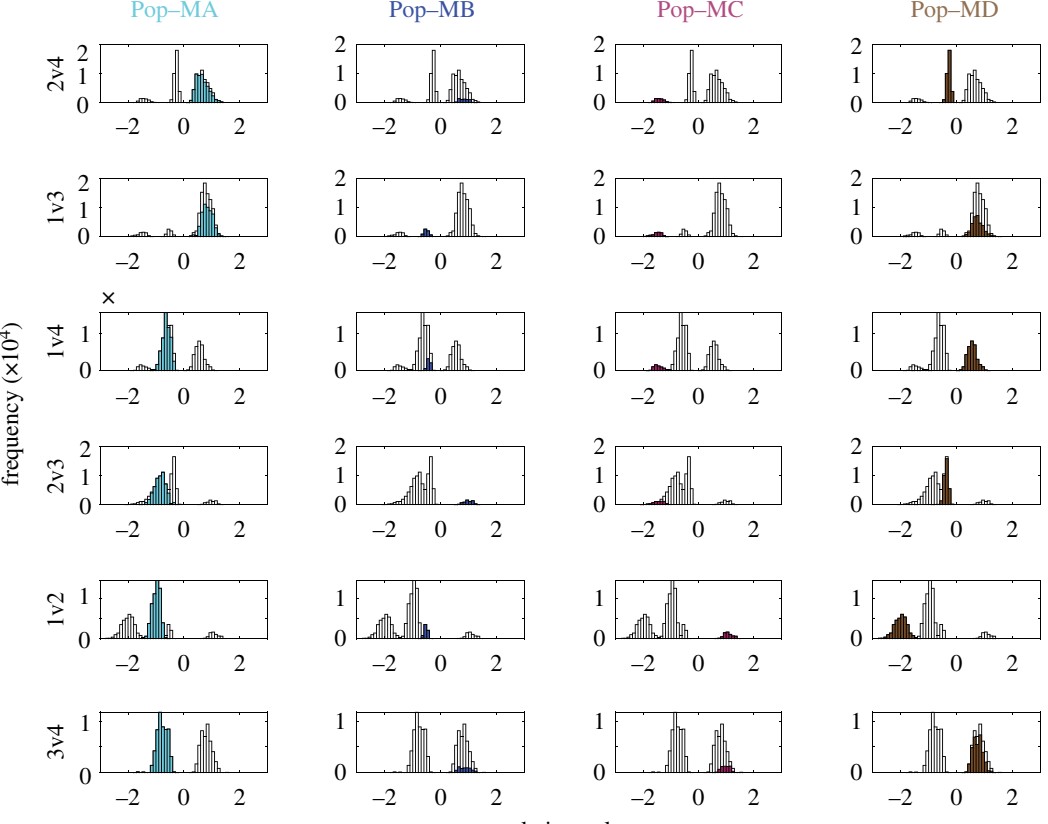

**Figure 4.** Four distinct, emergent population types (cryptic phenotypes) from model runs seeded with no bias. Each row displays one of the six possible trait correlations (2v4, 1v3, 1v4, 2v3, 1v2 and 3v4) with the distribution of the emergent trait correlation values for all individuals in all replicate runs ($n =$ 100 000) shown in grey. Highlighted in colour in each subplot are the trait correlation values for the individuals belonging to each of the emergent population types (columns). Each population type has a clearly defined set of trait correlation values. For example, the 2v4 mean correlation for Pop-MA was $0.66 \pm 0.22$ while the 2v4 mean correlation for pop-MD was $-0.28 \pm 0.07$. Pop-Ma and Pop-MD were the most accessible population types and so the trait correlation values associated with these population types (cryptic phenotypes) had the largest frequency (*y*-axis).

1 versus trait 2 (1v2), little to no overlap was observed across each of the four population types (figure 4, row 5), while for others, several population types shared the same trait correlations. For example, individuals in Pop-MA and Pop-MD shared the same 1v3 correlation (figure 4, row 2, columns 1 and 4) but had completely different relationships for 1v4 (figure 4, row 3, columns 1 and 4). The observance of emergent, cryptic phenotypes with distinct underlying trait combinations are qualitatively in line with experimental evolution studies that observed convergent phenotypes derived from a mix of parallel and divergent mutational and transcriptional changes across replicate populations adapting to the same environment [7,42–45].

When the model was run without the influence of trait correlations, only one phenotype emerged, as expected, but in contrast to the simulations where trait correlations were included (electronic supplementary material, figure S2 and information S2). Here individuals were unconstrained by bias and so were able to quickly move directly to the high-fitness area. This demonstrates that trait correlational constraints can produce different evolutionary strategies (i.e. emergent, cryptic phenotypes) and, if constraints are not present, individuals are able to explore phenotypic space more freely and arrive at the high-fitness phenotype more rapidly. The emergence of multiple high-fitness phenotypes (e.g. figure 5a) occupying a single high-fitness area in multivariate space demonstrates that, by using PCA for the trait-scape, our model captures

a rugged trait-based fitness landscape with multiple high-fitness peaks (e.g. figure 1).

Not all cryptic phenotypes were equally accessible by the model populations. Here, we define accessibility as the fraction of replicates that converged on an emergent population type. When the model was run without bias, Pop-MA was the most accessible with 55% of replicates converging on this population type while Pop-MD was the second most accessible with 33% (figure 5b). Pop-MA also exhibited the most variance in trait values within the population (i.e. broadest peak when plotted in more traditional pairwise trait space; e.g. Figure 5a), indicating a relatively larger range of trait values conferring high-fitness when associated with Pop-MA's trait correlations. The most accessible population type, Pop-MA, also had the fastest rate of fitness gain (figure 5b,c). Although Pop-MA and Pop-MB exhibited similar rates of adaptation (figure 5c, left plot), Pop-MB was not nearly as accessible with only 6% of the replicates converging on this population type (figure 5b). Instead, Pop-MD with a slower adaptive rate was the second most accessible population type (figure 5b,c). Pop-MA and Pop-MD trait correlations were more similar overall than those of Pop-MB.

To examine the impact of ancestral trait values on the accessibility of different cryptic phenotypes, we ran the mixed-mode model using (i) a single population at a different starting location (i.e. trait values) in the trait-scape that was equidistant to the high–fitness area and (ii) four different subpopulations

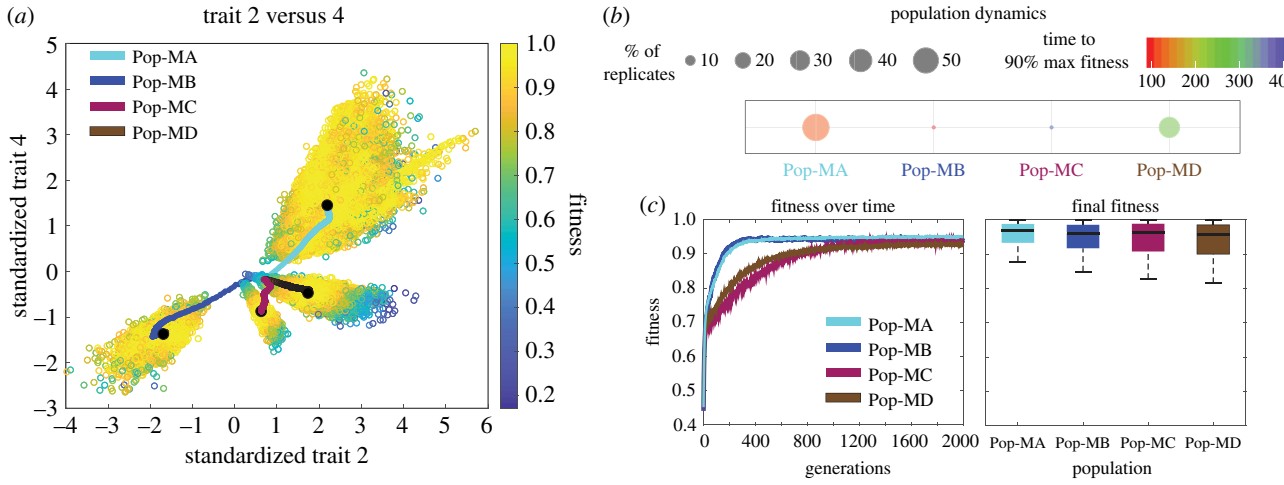

**Figure 5.** Representative trait, population and fitness dynamics in the TRACE model. (*a*) Trait versus trait plot denoting the four distinct population types (i.e. cryptic phenotypes) that emerged from 100 replicate model runs in mixed mode (i.e. no bias) with default model dynamics (90/10). Each hollow point represents the final trait values of a given individual in the last generation (2000th generation) coloured by fitness. Coloured lines represent the average trait values at each generation for each population with the black point denoting the final generation. (*b*) Population dynamics of the four emergent populations showing the number of replicates (out of 100) that resulted in specific populations (size of circle) along with each population's rate of adaptation (colour of circles). (*c*) The left plot displays the fitness of each population over time while the right displays boxplots representing the distribution of the final fitness values across all individuals of all replicate runs (*n* = 100 000). Black lines in the boxplots denote the median with the edges denoting the 25th and 75th percentiles.

(i.e. four different starting locations in the trait-scape). These model runs converged on the same cryptic phenotypes observed with the empirical starting location (Pop-MA, Pop-MB and Pop-MD for run 1 and all four cryptic phenotypes for run 2). However, shifting the starting location did alter the accessibility of the cryptic phenotypes (electronic supplementary material, figures S3 and S4). These runs indicate that the high-fitness phenotypes were conserved, and that starting an adaptive walk from another location influenced the accessibility of certain cryptic phenotypes thereby biasing evolutionary outcomes. The fact that no new populations emerged further supports the ability of this framework to capture the known phenomenon that there are a limited number of accessible phenotypes [27].

## (c) Adding historical bias

Movement of the populations within the trait-scape was impacted by the historical bias of the ancestral population (i.e. trait correlations). In the mixed-mode model, the population contained a large diversity of trait correlations among individuals with analogous trait values. The robustness of the resulting cryptic phenotypes across model runs indicates that certain trait correlations confirm a fitness advantage within the empirically defined trait-scape and thus were selected for. To assess how different types of bias (ancestral versus evolved correlations) impacted the accessibility of the cryptic phenotypes and rate of adaptation, we conducted a suite of simulations where bias was systematically added (sub-modes A1–A4 and sub-modes E1–E4). For both ancestral and evolved modes, systematically adding more bias (e.g. going from A1 to A4) changed the accessibility of the high-fitness phenotypes across replicate runs (electronic supplementary material, figure S5). However, the type of bias (e.g. ancestral versus evolved correlations) had a differential impact on cryptic phenotype accessibility. Bias from the ancestral correlations was typically maladaptive and resulted in fewer accessible cryptic phenotypes and slower adaptive rates (electronic supplementary material, figure S5a). On the

other hand, introducing bias derived from the observed evolved trait relationships (i.e. consistent with the trait-scape) generally resulted in faster adaptive rates and greater overall accessibility to the cryptic phenotypes (electronic supplementary material, figure S5b). These results are consistent with prior empirical and theoretical observations in developmental and quantitative genetic studies where bias (e.g. trait correlations) accelerated adaptive evolution if existing biological orientation aligned with the direction of selection but constrained adaptation if it limited variability in the direction of selection [16,17,27]. Specifically, depending on a starting population's bias, different phenotypes are more probable than others with some being generally inaccessible as found in other studies [27,28]. Here, we demonstrate these dynamics using a novel framework for modelling multivariate adaption in phytoplankton based on easy to quantify empirical trait data.

## (d) Meta-analysis of phenotypes across different modelled modes

We assessed the similarity between emergent cryptic phenotypes across all model simulations (9 modes with 100 replicates each) and showed five high-confidence cryptic phenotypes. Specifically, we grouped similar population types using hierarchical clustering with multiscale bootstrap resampling (1000 replicates) on mean trait correlation values at the 2000th generation. We also included the empirical data from the ancestral and evolved populations in this analysis. Hierarchical clustering revealed five high-confidence clusters (I–V) harbouring 93% of the phenotypes (*n* = 26 of 28) with AU *p*-values greater than 75 (figure 6*a*). Two population types, Pop-MB and Pop-EB-E1, clustered with II and IV, respectively, albeit with less confidence relative to the high-confidence clusters. The empirical ancestral population did not fall within any of the high-confidence clusters, which is expected as the ancestral population was not well adapted in the evolved trait-scape. By contrast, the empirical evolved population

Proc. R. Soc. B 288: 20210940

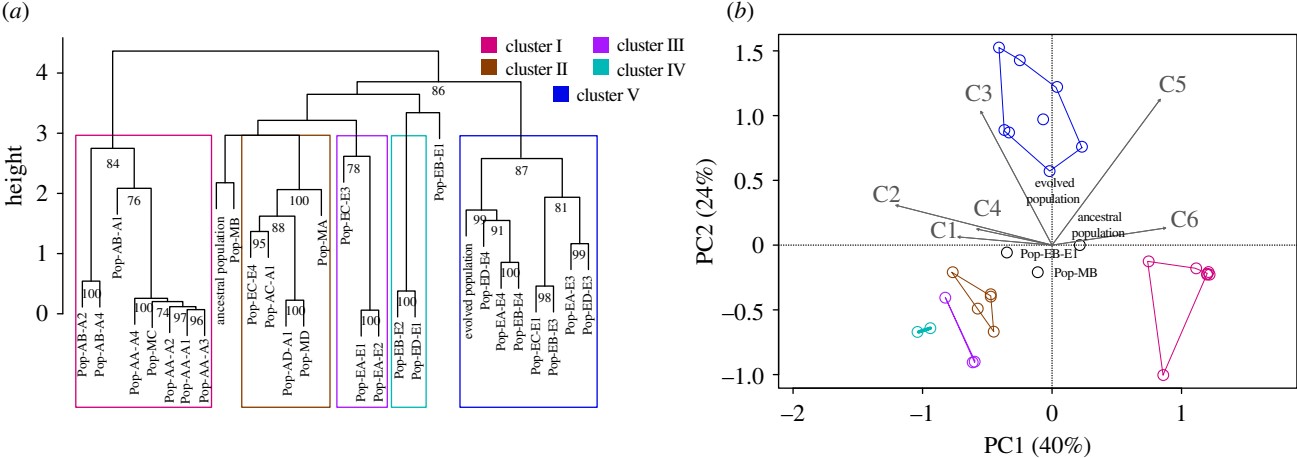

**Figure 6.** Hierarchical clustering and principal component analysis of mean trait correlation values calculated across all emergent population types from mixed, ancestral and evolved mode model simulations. (*a*) Hierarchical clustering with multiscale bootstrap resampling (1000 replicates) on trait correlation values from the emergent population types (e.g. figure 4) across all model runs (mixed mode, A1–A4 and E1–E4) along with empirical ancestral and evolved correlation values. AU *p*-values >75 are labelled at the nodes. We identified five overarching clusters with high-confidence AU *p*-values (colours), which contained even higher confidence sub-clusters. (*b*) Principal component analyses with trait correlation values as in (*a*) with the five clusters projected onto the coordinate plane as convex hulls. Percentages on *x*- and *y*-axis denote the per cent of explained variance along each axis. Vectors C1–C6 denote correlations 1–6 as defined in figure 2*c*.

clustered with high-confidence in cluster V. This empirically observed population type (cluster V) only emerged in the model simulation when evolved bias was present in the starting population and was found by 55% of the replicates from sub-modes E1, E3 and E4.

The clustering observed through the hierarchical analysis also emerged through a PC analysis of the population trait correlations. Specifically, we observed three general regions of convergence in PC space among the population types, as clusters II, III and IV collapsed into a small region of the lower left quadrant in the PCA plot (figure 6*b*). Importantly, these convergent regions emerged from thousands of possible trait and correlation values across varying degrees of bias. They provide valuable insight into probable combinations of high-$CO_2$ adaptive trait correlations along a reduced set of biological axes.

## 4. Discussion

We need to bridge the gap between evolutionary models and trait-based ecosystems models (ODE models) in order to better predict how marine microbes will adapt to shifts in the environment [11,46]. This work takes a critical first step in developing a framework (TRACE) which uses empirically derived multivariate trait-based landscapes to provide insight into the interaction between historical bias (trait correlations) and evolved phenotypes for marine phytoplankton. Critically, TRACE is derived from and provides predictions of easily quantifiable traits—such as those commonly measured by biological oceanographers. Using data from an experimental evolution study with a model green alga, we found that a limited set of integrated phenotypes underlie thousands of possible trait correlational scenarios and that only certain phenotypes were accessible depending on the amount and type of bias. By leveraging empirical ancestral trait correlations and the observed changes in these correlations as a result of adaptation to high $CO_2$, we were able to simulate adaptive walks with endpoints anchored in real evolutionary outcomes.

This study provides a roadmap for future integration of evolutionary theory with biological oceanography. While we focus on a case study using an experimental evolution study to generate the trait-scape, our framework can be used to generate a trait-scape for any given environment A using trait data from an *in situ* population (assuming that the *in situ* population is well adapted to environment A). This trait-scape could then be used with the TRACE framework to develop hypotheses as to how a new population from a different environment B might adapt upon exposure to environment A. This insight can be gained with easily quantifiable trait measurements and without requiring genetic manipulations, which are currently not possible for most marine microbes.

Our work demonstrates that ecological models need to represent both changes in traits (already existing in some ecological models) and changes in the correlation between traits in order to accurately capture phytoplankton evolution. The TRACE framework could be combined with an ODE based ecosystem model to predict marine microbial adaptation more accurately. While the development of a fully integrated TRACE + ODE model is beyond the scope of this paper, we demonstrate that such an integration would result in a substantially different set of phenotypes and impact community dynamics (electronic supplementary material, information S4 and figures S6 and S7). Such an integration provides the added benefit of removing the need to define the evolutionary endpoint, as fitness in an ODE model can be dynamically estimated based on prognostically calculated growth and mortality rates.

This study demonstrates that shifts in trait correlations are fundamental for understanding the evolved phenotype and provides a novel framework for linking easily quantifiable trait measurements to predictions of evolved phenotypes for marine phytoplankton. This is particularly exciting because we can create trait-scapes from field data and use TRACE to understand how invading microbial populations may be able to adapt (i.e. create new phenotypes). Importantly, this approach can also help inform future experimental designs aimed at testing the probability of adaptive outcomes across multivariate environments through the analysis of a select

set of traits. We are at a critical juncture where we need ecosystem and biogeochemical models to incorporate evolutionary dynamics in order to robustly predict future shifts in ecosystem dynamics [33]. Due to the seemingly infinite amount of possible interacting biological and environmental variables to test, evolutionary and mathematical tools that allow us to efficiently combine experiments with modelling will be critical to help predict microbial population responses to future global change scenarios through the lens of evolutionary phenomena.

Data accessibility. The model code is available at https://github.com/LevineLab and a version of the manuscript is available from the biology preprint server *bioRxiv*: https://www.biorxiv.org/content/10.1101/2020.08.04.237230 [47].

The data are provided in electronic supplementary material [48].

Authors' contributions. N.G.W.: conceptualization, data curation, formal analysis, investigation, methodology, project administration, software, validation, visualization, writing-original draft, writing-review and editing; J.H.: conceptualization, investigation, methodology, writing-review and editing; P.A.A.: investigation, methodology, writing-review and editing; S.G.L.: data curation, formal analysis, investigation, methodology, visualization, writing-review and editing; M.A.D.: conceptualization, funding acquisition, investigation, methodology, writing-review and editing; S.C.: conceptualization, data curation, formal analysis, funding acquisition, investigation, methodology, project administration, resources, supervision, validation, writing-review and editing; N.M.L.: conceptualization, data curation, formal analysis, funding acquisition, investigation, methodology, project administration, resources, software, supervision, validation, writing-original draft, writing-review and editing.

All authors gave final approval for publication and agreed to be held accountable for the work performed therein.

Competing interests. We declare that we have no competing interests.

Funding. This work was supported by the Moore Foundation grant MMI 7397 (to N.M.L., S.C., M.A.D.) and by the Simons Foundation grant 509727 (to N.M.L.).

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
