## [Peer Review File · Proceedings of the Royal Society B: Biological Sciences]

Review History

RSPB-2020-3144.R0 (Original submission)

Review form: Reviewer 1

Recommendation

Accept with minor revision (please list in comments)

Scientific importance: Is the manuscript an original and important contribution to its field?

Good

General interest: Is the paper of sufficient general interest?

Good

Quality of the paper: Is the overall quality of the paper suitable?

Good

Is the length of the paper justified?

Yes

Should the paper be seen by a specialist statistical reviewer?

No

Do you have any concerns about statistical analyses in this paper? If so, please specify them explicitly in your report.

No

It is a condition of publication that authors make their supporting data, code and materials available - either as supplementary material or hosted in an external repository. Please rate, if applicable, the supporting data on the following criteria.

Is it accessible?

Yes

Is it clear?

Yes

Is it adequate?

Yes

Do you have any ethical concerns with this paper?

No

Comments to the Author

This manuscript is concerned with multitrait variation and evolution. It examines ways in which evolving organisms might be constrained to access only a subset of available phenotype combinations in multitrait space. This is a worthwhile undertaking, concerned with processes that might limit the capacity of organisms to respond to change, and influence the biogeochemical cycles those organisms mediate.

The paper is excellent in most respects. The research outcomes are potentially interesting and important, and the manuscript is largely well-crafted. I had two comments, which are described in detail below. The first concerns the presentation of the model. I found it very hard to understand how the simulations worked, and recommend the addition of details, possibly to the supplement. The second is that I wondered about aspects of the rationale for evolving trait correlations, and wondered if this could be clarified.

1. Presentation of the model.

I suggest adding more detail about what happens in each simulation, possibly in the supplement. What happens as it is stepped forward through generations? Are some individuals reproducing and others being lost according to fitness functions linked to current trait values? Or are they only experiencing changes in their trait values? (If this information was supplied and I missed it, my apologies.) I suspect these details are probably available in previous articles by the authors and others. However, I think it would be helpful to repeat it here, for two reasons. First, these details might bring assumptions that are worth stating explicitly. For example, the simulations probably assume steady state population dynamics, which presumably affect the fitness consequences of different traits, and potentially, combinations of traits. It seems like it would be important that the dynamics of the simulations match the data that inform their parameterization, or is this not necessary? I was particularly concerned about this because the results are discussed in the context of biogeochemical models, which often have regimes of both steady state conditions and highly episodic boom-bust growth conditions. Second, one of the traits that is examined is growth rate, which is an important parameter in many forms of microbial population models. I was initially confused that the growth rate data were being used to parameterize statistical models of trait change, rather than population dynamics of different lineages in a simulation. Possibly this will also confuse others that are more familiar with ODE models of microbe population dynamics, so I suggest it might be worthwhile to be very explicit about what is happening in the simulations.

2. The evolution of trait correlations.

In the case of historical bias, I was curious about the rationale for imposing initial conditions of trait correlation values, but then allowing those values to take a random walk through the simulations. In broader applications (e.g. involving nutrient processing, temperature responses) key traits sometimes have strong tradeoffs. Is there a way to use the data to infer when there is justification for setting not only initial trait correlation values, but also constraining how much they can vary? Otherwise, over a sufficiently long simulation, it seems like organisms could evolve to have trait combinations that are functionally very unlikely (on a physical or chemical basis).

In sum, the article has the potential to make a useful contribution to the literature, and provides insights about how adaptation could be integrated into biogeochemical models. But the paper attempts to make quite a big conceptual link, between a statistical model for trait variation, and biogeochemical models where those traits could be used to inform condition-dependent parameters for transformations. Some of the contact points between those two things are not clear at present, and might be clarified by details of the simulations.

Review form: Reviewer 2

Recommendation

Reject – article is not of sufficient interest (we will consider a transfer to another journal)

Scientific importance: Is the manuscript an original and important contribution to its field?

Marginal

General interest: Is the paper of sufficient general interest?

Marginal

Quality of the paper: Is the overall quality of the paper suitable?

Marginal

Is the length of the paper justified?

Yes

Should the paper be seen by a specialist statistical reviewer?

No

Do you have any concerns about statistical analyses in this paper? If so, please specify them explicitly in your report.

No

It is a condition of publication that authors make their supporting data, code and materials available - either as supplementary material or hosted in an external repository. Please rate, if applicable, the supporting data on the following criteria.

Is it accessible?

Yes

Is it clear?

No

Is it adequate?

Yes

Do you have any ethical concerns with this paper?

No

Comments to the Author

I'm not sure I fully understand the point of this paper - we know that trait correlations and initial phenotypes matter for evolutionary trajectories more generally - this is specific example of that but I'm not sure what we've learned generally from it. I don't have any problem with work per se, it's just the paper doesn't make it clear what new things we learn from this exercise. The conclusions are not different to what conclusions I would draw from reading say, Lynch and Walsh. I'm sorry I can't be more supportive, it's just I really don't know what is being argued here. The authors use a specific example, but then regard the traits themselves as generic, so I don't really see how the two things connect. If there's some deeper, novel point here, I'm sorry to say that it is beyond me.

Decision letter (RSPB-2020-3144.R0)

15-Feb-2021

Dear Professor Levine:

I am writing to inform you that your manuscript RSPB-2020-3144 entitled "The evolution of trait correlations constrains phenotypic adaptation to high CO₂ in a eukaryotic alga" has, in its current form, been rejected for publication in Proceedings B.

This action has been taken on the advice of referees, who have recommended that substantial revisions are necessary. With this in mind we would be happy to consider a resubmission, provided the comments of the referees are fully addressed. However please note that this is not a provisional acceptance.

Sincerely,
Dr Maurine Neiman
mailto:proceedingsb@royalsociety.org

Associate Editor
Board Member: 1
Comments to Author:

Thank you for submitting your manuscript to Proceedings B. Your manuscript has now been reviewed by two expert reviewers and myself. You will see that the reviews were mixed. Reviewer 2 did not see the novelty in the manuscript or how the findings presented could reasonably be generalized beyond the case presented. Reviewer 1 was more positive, finding the manuscript mostly well-written and potentially important, but also found it lacking in making a link between the specific simulation of trait variation and a more general inference about fitness within a physiological and biogeochemical context. Reviewer 1 also had criticisms about the description of the methods and choices made in designing the simulation. In summary, there is a disconnect between what the authors see as the implications and impact of this work and what is communicated by the manuscript.

Based on these reviews and my own evaluation, my recommendation is to reject with the option to resubmit. A revised manuscript would have to address each of the reviewer criticisms. The specific concerns about the presentation of the model and the evaluation of trait correlations raised by Reviewer 1 would need to be clearly and explicitly addressed. The more general concerns of both reviewers would also have to be satisfactorily dealt with: that the conclusions are not generalizable from this one case study, that the link between trait correlations and more general biogeochemical models is unsupported, and that therefore the work does not present a significant advance. Being mindful of the broad readership of Proceedings B, the narrative should be as accessible as possible to biologists and ecologists who are not modelers. If you choose to revise and resubmit, these changes should be reflected in the manuscript and in your point-by-point response to the reviewers.

Reviewer(s)' Comments to Author:
Referee: 1

Comments to the Author(s)

This manuscript is concerned with multitrait variation and evolution. It examines ways in which evolving organisms might be constrained to access only a subset of available phenotype combinations in multitrait space. This is a worthwhile undertaking, concerned with processes that might limit the capacity of organisms to respond to change, and influence the biogeochemical cycles those organisms mediate.

The paper is excellent in most respects. The research outcomes are potentially interesting and important, and the manuscript is largely well-crafted. I had two comments, which are described in detail below. The first concerns the presentation of the model. I found it very hard to understand how the simulations worked, and recommend the addition of details, possibly to the supplement. The second is that I wondered about aspects of the rationale for evolving trait correlations, and wondered if this could be clarified.

1. Presentation of the model.

I suggest adding more detail about what happens in each simulation, possibly in the supplement. What happens as it is stepped forward through generations? Are some individuals reproducing and others being lost according to fitness functions linked to current trait values? Or are they only experiencing changes in their trait values? (If this information was supplied and I missed it, my apologies.) I suspect these details are probably available in previous articles by the authors and others. However, I think it would be helpful to repeat it here, for two reasons. First, these details might bring assumptions that are worth stating explicitly. For example, the simulations probably assume steady state population dynamics, which presumably affect the fitness consequences of

different traits, and potentially, combinations of traits. It seems like it would be important that the dynamics of the simulations match the data that inform their parameterization, or is this not necessary? I was particularly concerned about this because the results are discussed in the context of biogeochemical models, which often have regimes of both steady state conditions and highly episodic boom-bust growth conditions. Second, one of the traits that is examined is growth rate, which is an important parameter in many forms of microbial population models. I was initially confused that the growth rate data were being used to parameterize statistical models of trait change, rather than population dynamics of different lineages in a simulation. Possibly this will also confuse others that are more familiar with ODE models of microbe population dynamics, so I suggest it might be worthwhile to be very explicit about what is happening in the simulations.

2. The evolution of trait correlations.

In the case of historical bias, I was curious about the rationale for imposing initial conditions of trait correlation values, but then allowing those values to take a random walk through the simulations. In broader applications (e.g. involving nutrient processing, temperature responses) key traits sometimes have strong tradeoffs. Is there a way to use the data to infer when there is justification for setting not only initial trait correlation values, but also constraining how much they can vary? Otherwise, over a sufficiently long simulation, it seems like organisms could evolve to have trait combinations that are functionally very unlikely (on a physical or chemical basis).

In sum, the article has the potential to make a useful contribution to the literature, and provides insights about how adaptation could be integrated into biogeochemical models. But the paper attempts to make quite a big conceptual link, between a statistical model for trait variation, and biogeochemical models where those traits could be used to inform condition-dependent parameters for transformations. Some of the contact points between those two things are not clear at present, and might be clarified by details of the simulations.

Referee: 2

Comments to the Author(s)

I'm not sure I fully understand the point of this paper - we know that trait correlations and initial phenotypes matter for evolutionary trajectories more generally - this is specific example of that but I'm not sure what we've learned generally from it. I don't have any problem with work per se, it's just the paper doesn't make it clear what new things we learn from this exercise. The conclusions are not different to what conclusions I would draw from reading say, Lynch and Walsh. I'm sorry I can't be more supportive, it's just I really don't know what is being argued here. The authors use a specific example, but then regard the traits themselves as generic, so I don't really see how the two things connect. If there's some deeper, novel point here, I'm sorry to say that it is beyond me.

Author's Response to Decision Letter for (RSPB-2020-3144.R0)

See Appendix A.

RSPB-2021-0940.R0

Review form: Reviewer 2

Recommendation

Accept as is

Scientific importance: Is the manuscript an original and important contribution to its field?

Marginal

General interest: Is the paper of sufficient general interest?

Poor

Quality of the paper: Is the overall quality of the paper suitable?

Good

Is the length of the paper justified?

Yes

Should the paper be seen by a specialist statistical reviewer?

No

Do you have any concerns about statistical analyses in this paper? If so, please specify them explicitly in your report.

No

It is a condition of publication that authors make their supporting data, code and materials available - either as supplementary material or hosted in an external repository. Please rate, if applicable, the supporting data on the following criteria.

Is it accessible?

Yes

Is it clear?

Yes

Is it adequate?

Yes

Do you have any ethical concerns with this paper?

No

Comments to the Author

Based on the authors responses, I'm not convinced that this journal is the best venue for this work - the findings aren't novel to an evolution crowd and would seem to me to have much more potential for impact if aimed at a biogeochemistry crowd. But given the work was solicited from a preprint, I don't think this is my call to make. I have no specific comments about the ms as I think it's fine work.

Decision letter (RSPB-2021-0940.R0)

07-May-2021

Dear Professor Levine

I am pleased to inform you that your Review manuscript RSPB-2021-0940 entitled "The evolution of trait correlations constrains phenotypic adaptation to high CO₂ in a eukaryotic alga" has been accepted for publication in Proceedings B.

The referee(s) do not recommend any further changes. Therefore, please proof-read your manuscript carefully and upload your final files for publication. Because the schedule for publication is very tight, it is a condition of publication that you submit the revised version of your manuscript within 7 days. If you do not think you will be able to meet this date please let me know immediately.

To upload your manuscript, log into <http://mc.manuscriptcentral.com/prsb> and enter your Author Centre, where you will find your manuscript title listed under "Manuscripts with Decisions." Under "Actions," click on "Create a Revision." Your manuscript number has been appended to denote a revision.

You will be unable to make your revisions on the originally submitted version of the manuscript. Instead, upload a new version through your Author Centre.

- 1) A text file of the manuscript (doc, txt, rtf or tex), including the references, tables (including captions) and figure captions. Please remove any tracked changes from the text before submission. PDF files are not an accepted format for the "Main Document".
- 2) A separate electronic file of each figure (tiff, EPS or print-quality PDF preferred). The format should be produced directly from original creation package, or original software format. Please note that PowerPoint files are not accepted.
- 3) Electronic supplementary material: this should be contained in a separate file from the main text and the file name should contain the author's name and journal name, e.g. `authorname_procb_ESM_figures.pdf`

All supplementary materials accompanying an accepted article will be treated as in their final form. They will be published alongside the paper on the journal website and posted on the online figshare repository. Files on figshare will be made available approximately one week before the accompanying article so that the supplementary material can be attributed a unique DOI. Please see: <https://royalsociety.org/journals/authors/author-guidelines/>

4) Data-Sharing and data citation

It is a condition of publication that data supporting your paper are made available. Data should be made available either in the electronic supplementary material or through an appropriate repository. Details of how to access data should be included in your paper. Please see <https://royalsociety.org/journals/ethics-policies/data-sharing-mining/> for more details.

<http://datadryad.org/submit?journalID=RSPB&manu=RSPB-2021-0940> which will take you to your unique entry in the Dryad repository.

Once again, thank you for submitting your manuscript to Proceedings B and I look forward to receiving your final version. If you have any questions at all, please do not hesitate to get in touch.

Sincerely,

Dr Maurine Neiman
<mailto:proceedingsb@royalsociety.org>

Associate Editor

Comments to Author:

Thank you for your revisions, which are comprehensive. There are no further revisions requested by the reviewers or associate editor.

Reviewer(s)' Comments to Author:

Referee: 2

Comments to the Author(s).

Based on the authors responses, I'm not convinced that this journal is the best venue for this work - the findings aren't novel to an evolution crowd and would seem to me to have much more potential for impact if aimed at a biogeochemistry crowd. But given the work was solicited from a preprint, I don't think this is my call to make. I have no specific comments about the ms as I think it's fine work.

Decision letter (RSPB-2021-0940.R1)

17-May-2021

Dear Professor Levine

I am pleased to inform you that your manuscript entitled "The evolution of trait correlations constrains phenotypic adaptation to high CO₂ in a eukaryotic alga" has been accepted for publication in Proceedings B.

Data Accessibility section

Open Access

Paper charges

Sincerely,

Proceedings B

Appendix A

April 21st, 2021

Dear Editor:

We are **resubmitting** our manuscript titled “**The evolution of trait correlations constrains phenotypic adaptation to high CO₂ in a eukaryotic alga**” for publication as an Article in *Proceedings of the Royal Society B*. The original submission (RSPB-2020-3144) was solicited by Dr. Maurine Neiman of the Proceedings B Preprint Editorial Team who found our paper on BioRxiv and encouraged us to submit to *Proc B*. We have carefully noted your comments and suggestions, and the Reviewers’ comments and suggestions, in producing this revised version of the manuscript. We think these comments have helped to further clarify our results and strengthen the paper. Specifically, we have substantially revised the main text, including rewriting the entire discussion, to highlight the novelty of our work. We have conducted a new model analysis to demonstrate the ability of our approach to bridge evolutionary models and ODE-based biogeochemical models. Finally, we have run additional sensitivity analyses and expanded the description of our model. We detail these changes in the point-by-point response to the Reviewers’ comments.

Microbes form the base of global biosphere and drive both aquatic and terrestrial biogeochemical cycling, thereby significantly influencing Earth’s food webs and climate. Predicting how microbial populations adapt and how this will influence global elemental cycles remains a fundamental challenge. We are at a critical juncture where we need ecosystem and biogeochemical models to incorporate evolutionary dynamics in order to robustly predict future shifts in ecosystem dynamics. The primary aim of this paper was to create a bridge between these two disciplines by developing a novel framework based on easily quantifiable microbial traits. We show that this new framework is consistent with existing evolutionary theory and can be used to understand constraints on quantifiable trait-based phenotypes such as those measured in the laboratory and field and modeled by biogeochemists. We thus have provided a framework which can be used with easily obtained trait data to understand constraints on phenotype adaptation for globally relevant microbes. An additional major advance in our approach is that it can be used on data from unculturable organisms and is scalable to a large number of traits. Our work further demonstrates that ecological models need to represent both changes in traits (already existing in some ecological models) and changes in the correlation between traits in order to accurately capture phytoplankton evolution – and proposes a framework for such an integration. We believe our findings are of interest to widespread scientific fields including evolutionary biology, physiology, biological and chemical oceanography, aquatic ecology, biogeochemistry, and ecosystem modeling.

The manuscript is 4,116 words long and contains 6 display items and 2006 words of additional supplementary text with 2 supplementary tables, 1 supplementary file, and 9 supplementary figures.

Thank you in advance for your time.

Sincerely,

Naomi M. Levine

Gabilan Assistant Professor of Biological Sciences and Earth Sciences

Nathan Walworth

Postdoctoral Scholar of Biological Sciences

**Reviewer Responses**

**Reviewer 1**

*1. Presentation of the model.*

*1a. I suggest adding more detail about what happens in each simulation, possibly in the*
*supplement. What happens as it is stepped forward through generations? Are some individuals*
*reproducing and others being lost according to fitness functions linked to current trait values? Or*
*are they only experiencing changes in their trait values? (If this information was supplied and I*
*missed it, my apologies.)*

**Response:** We appreciate the reviewer's request for further clarity here. We have modified the
text to clarify the model dynamics (revised text: L 158 – 164, L 175). Given length constraints, we
have primarily expanded the description in the supplement (Supplemental Information S3). A brief
description of the model dynamics is also provided below in response to comment 1b.

*1b. I suspect these details are probably available in previous articles by the authors and others.*
*However, I think it would be helpful to repeat it here, for two reasons. First, these details might*
*bring assumptions that are worth stating explicitly. For example, the simulations probably assume*
*steady state population dynamics, which presumably affect the fitness consequences of different*
*traits, and potentially, combinations of traits. It seems like it would be important that the dynamics*
*of the simulations match the data that inform their parameterization, or is this not necessary?*

**Response:** We thank the reviewer for these helpful comments on how to clarify the
presentation of our model framework. We have revised the text to clarify that the model represents
the adaption of a population of 1000 individuals that experience random changes in trait values
and trait correlations. At each generation, the individuals with the highest fitness are more likely
to persist and create 'offspring'. Fitness in the model is a function of difference between the
individuals' trait values and the trait values at the optimum (high-fitness phenotype). Please see
above references to line numbers above where we updated the main and supplementary
information S3.

We assume a constant population size (1000 individuals) over the entire simulation. We
extensively tested the impact of different population sizes with previous versions of this model
(Kronholm and Collins 2015; Walworth et al. 2020) and showed that results were consistent
across a large range of population sizes with the same mutational supply. Selection timescales
depend on the neutral mutation rate (μ) and the effective population size (N_e) rather than census
population size. The mutational supply for a population can be represented by the composite
parameter $N_e\mu$ (Hartl and Taubes 1998). Kronholm and Collins, 2015 tested a range of values of
$N_e\mu$ and found that the results were similar within the range of plausible values where drift could
be overcome (ie – when N_e was not extremely small).

As TRACE uses an adaptive random walk to capture evolutionary dynamics, we are not
directly simulating population dynamics the way an ODE based ecosystem model would. A more
detailed comparison between our model and typical ODE ecosystems models is provided below in
response to comment 1c.

*1c. I was particularly concerned about this because the results are discussed in the context of*
*biogeochemical models, which often have regimes of both steady state conditions and highly*
*episodic boom-bust growth conditions. Second, one of the traits that is examined is growth rate,*
*which is an important parameter in many forms of microbial population models. I was initially*
*confused that the growth rate data were being used to parameterize statistical models of trait*

*change, rather than population dynamics of different lineages in a simulation. Possibly this will*
*also confuse others that are more familiar with ODE models of microbe population dynamics, so*
*I suggest it might be worthwhile to be very explicit about what is happening in the simulations.*

**Response:** This highlights an important point which we have tried to clarify in the revised text.
The TRACE model presented in this paper captures shifts in traits and trait-correlations that occur
as phytoplankton adapt to a new environment. TRACE is a novel framework which allows us to
link evolutionary theory of how adaptation proceeds with quantifiable microbial traits. In this
paper, we demonstrate that correlations between traits in the ancestral populations (akin to
historical effects on adaptation, or genetic constraints) constrain the adaptive walk of the
population and influence the evolved phenotypes. To ground our model in observations, we use
observed shifts in traits and trait relationship (quantified using PCA) from an experimental
evolution study with *Chlamydomonas*. Growth rate was one of the traits quantified in the
experiment which not only shifted significantly between the ancestral and evolved populations but
also changed in relationship to other quantified traits. As such, we selected it as one of the traits to
simulate in TRACE – however, TRACE is agnostic as to what the actual traits are. Because the
dynamics within TRACE are not linked to the growth dynamics of the individuals, including
growth rate as a trait is justified. We agree that this could create some confusion and so have tried
to clarify this point in the text and refer to the traits simply as traits 1-4 to help minimize confusion.

An additional reason for selecting growth rate as a trait of interest is that it is a commonly
measured trait in both laboratory and field based studies. As such, we felt including growth rate
might resonate with other researchers who might want to use TRACE to better understand their
system. Since growth rate is typically used as a key *proxy* for fitness in microbial populations, we
believe it is important to examine how it relates to other ecologically important traits in a
multivariate landscape, or the integrated phenotype. However, it is worth noting that, in the context
of an evolutionary model, fitness is a relative value. Specifically, fitness is the relative reproductive
success of a focal genotype, in the environment of interest, against other genotypes that are present
– as expected, growth rate in the absence of conspecific competitors is an important component of
fitness (Wolf et al. 2019). We have shown before that competitive value and pure culture growth
rates need not be equivalent in microalgae (Collins and Schaum 2019).

One of the key challenges in understanding how phytoplankton will adapt to shifts in the
environments is that, at present, evolutionary models are not suited to interface with
biogeochemical ODE models (Ward et al. 2019). One of our aims in developing TRACE was to
bring evolutionary models and trait-based ecosystems models (ODE models) closer together.
While TRACE is not a mechanistic model (e.g. growth rate is not solved for prognostically), it
provides important insight into how trait adaptation might be implemented in ODE models.
Specifically, the trait-correlations in TRACE are analogous to the allometries prescribed in ODE
models (e.g. minimum nitrogen quota vs size). A population in an ODE model will respond to
changes in environment with shifts in trait values as a result of these allometries (captured as trait
changes in TRACE). While previous studies have looked at adaptation in ODE models by
‘mutating’ parameter values such as cell size (e.g. Sauterey et al. 2015; Sauterey et al. 2017; Jiang
et al. 2005; Nakov et al. 2014; Kremer and Klausmeier 2013), maximum growth rate (Taherzadeh
et al. 2017), and optimum temperature (Beckmann et al 2019), to our knowledge, evolutionary
changes in trait correlations have not yet been included in ODE models – instead, trait correlations
are treated as fundamental tradeoffs that cannot change (evolve). However, we know that shifts
in trait-correlations do occur (Lindberg & Collins 2020; Malerba and Marshall 2019) and this study
shows that these shifts are fundamental for understanding the evolved phenotypes. TRACE

provides a framework for evolving the relationships between traits, as well as trait values
themselves, thus allowing us to relax the assumption in ODE models that trait correlations are
fixed in populations responding to environmental change. Finally, while TRACE requires the
definition of an evolutionary end point, once this framework is integrated into an ODE model,
fitness can be dynamically estimated based on prognostically calculated growth and mortality
rates. While the development of a fully integrated TRACE + ODE model is beyond the scope of
this paper, we have added a supplemental section (see Supplementary information S4: *Integrating*
*the TRACE approach into an ODE framework*) which includes a set of simulations that
demonstrate the power of TRACE to link evolutionary theory to classic microbial ecosystem
models. We have also added in a more detailed explanation of how TRACE can be expanded to
interface with conventional ODE models of population dynamics (see newly written *Discussion*
section).

*2. The evolution of trait correlations.*

*In the case of historical bias, I was curious about the rationale for imposing initial conditions of*
*trait correlation values, but then allowing those values to take a random walk through the*
*simulations. In broader applications (e.g. involving nutrient processing, temperature responses)*
*key traits sometimes have strong tradeoffs. Is there a way to use the data to infer when there is*
*justification for setting not only initial trait correlation values, but also constraining how much*
*they can vary? Otherwise, over a sufficiently long simulation, it seems like organisms could evolve*
*to have trait combinations that are functionally very unlikely (on a physical or chemical basis).*

**Response:** We thank the Reviewer for highlighting this point. We completely agree with the
Reviewer's comment and feel that this is a key aspect of our paper – the inherent trait correlations
present in an initial population play a significant role in determining the adaptive outcome. The
trait-trade-offs that the reviewer refers to are equivalent to the trait-correlations represented in
TRACE. Previous empirical work has shown that these relationships can (and do) evolve
(Lindberg & Collins 2020; Malerba et al. 2017). In the model, populations move within the 'trait-
scape' from the ancestral origin towards a high fitness optimum. The movement within the trait-
scape is determined by the trait-correlations of the individuals. Both the trait-scape and the high
fitness optimum are derived directly from empirical data. As such, the model is selecting for the
observed trait combinations. While the high fitness optimum is only defined in terms of trait
values, because movement is controlled by trait correlations, we are also indirectly selecting for
populations with trait-correlations that are consistent with the observed relationships. For
example, take two individuals A and B with identical trait values but different trait-correlations.
Let us suppose that both individual A and B experience a change in trait 1 that would move them
in the trait-scape closer to the optimum. However, because trait 1 is linked to the other traits, the
change in trait 1 (which is identical in individual A and B) will result in changes to traits 2-N that
will not be identical in individual A and B because of the different trait-correlations. As a result,
A and B will not have identical movements in the trait-scape and one most likely will be closer to
the high-fitness optimum and therefore more likely to persist. What is exciting about our approach,
is that it provides insight into how the trait-correlations present in the initial population impact the
adaptive outcome. This highlights the importance of including both changes in trait values and
changes in trait-correlations into marine ecosystem models that represent adaptation. Furthermore,
because TRACE is parameterized using high-fitness trait combinations derived from actual
observations, there is not a concern that the model will generate organisms with trait-combinations
that are not physically or chemically possible. We have modified the discussion to highlight this

point and have added in additional simulations using an ODE based ecosystem model to reinforce
this point (see above).

The reviewer's comment brings up an additional point – while we vary the 'genetic diversity'
in our starting population (variable trait-correlation), we chose to seed the model with a single
phenotype (trait values). Specifically, we selected the starting point based on the observed trait
values from the experimental evolution study. It is important to note that while all individuals
started with the same phenotype, they were not a true clonal population in the model as the trait-
correlations varied between individuals for most of our simulations. However, the reviewer brings
up an important point that relates to one of the key findings of our work – often there are multiple
high-fitness trait combinations for a given environment. To address this, we have conducted an
additional set of simulations in which we initialize the model with a diverse population both in
terms of trait-correlations and in terms of trait values (phenotypes). To link these additional
simulations with the empirical data, we use 4 different phenotypes that were present in the ancestral
data of the Lindberg & Collins (2020) experiment. These runs converged on the same 4 emergent
phenotypes as in Fig. 4 indicating that the phenotypic diversity of the initial population does not
impact our results. Due to space constraints, the figures for these additional simulations are
included in the supplement (Supplementary Fig. 1) but are referred to on lines 269 – 273 of the
revised text.

**Reviewer 2**

*I'm not sure I fully understand the point of this paper - we know that trait correlations and initial*
*phenotypes matter for evolutionary trajectories more generally - this is specific example of that*
*but I'm not sure what we've learned generally from it. I don't have any problem with work per se,*
*it's just the paper doesn't make it clear what new things we learn from this exercise. The*
*conclusions are not different to what conclusions I would draw from reading say, Lynch and*
*Walsh. I'm sorry I can't be more supportive, it's just I really don't know what is being argued here.*
*The authors use a specific example, but then regard the traits themselves as generic, so I don't*
*really see how the two things connect. If there's some deeper, novel point here, I'm sorry to say*
*that it is beyond me.*

**Response:** We thank the reviewer for highlighting the need to clarify the novelty in our approach
in adapting evolutionary modeling to an ecosystem framework. We have substantially edited the
text to clarify the unique contribution that we are making and have added in several additional
analyses to highlight the novelty. Below we describe the novelties in our approach in terms of
bridging evolutionary and ODE-based biogeochemical models and creating fitness landscapes
using easily quantifiable traits.

*Bridging evolutionary and ODE-based biogeochemical models*

This work demonstrates that we can use fitness landscapes derived directly from empirical trait
values to capture known evolutionary phenomena. This is particularly exciting because we can use
TRACE with field data to understand how invading microbial populations may be able to adapt
(i.e. create new phenotypes) in response to ecological and environmental change. This capability
is currently non-existent in biogeochemical models. Most important, to our knowledge, all
previous studies on phytoplankton adaptation have focused on changes in trait values (e.g. cell size
or maximum growth rate) and have not considered the importance of the relationships between
traits in determine the adaptive phenotypes. ***The novelty in our work lies in the creation of a***
***bridge between evolutionary models and ODE based biogeochemical models.*** While the field of
evolution has made great strides in the past few decades, the gulf between evolution literature and
biological oceanography has widened such that current state-of-the-art evolutionary models are
not well suited to interface with biogeochemical ODE models (Ward et al. 2019). We are at a
critical juncture where we need ecosystem and biogeochemical models to incorporate evolutionary
dynamics in order to robustly predict future shifts in ecosystem dynamics (Baltar et al. 2019).
Thus, the primary aim of this paper was to create a bridge between the two fields by developing a
framework which uses evolutionary theory to generate trait-correlation outcomes that can be
validated with and used to predict quantifiable trait-based phenotypes such as those measured in
the laboratory and field and modeled by biogeochemists. We were inspired by the work of Chevin
et al. (2010) who used an evolutionary model and environmental tolerance curves to explore major
determinants of extinction risk to environmental change. Chevin et al's approach was not
necessarily novel in terms of theory but produced a description of the relationship between
plasticity and evolution that largely relied on parameters that could be estimated empirically, and
a model that could be tested empirically, which allowed the body of theory to actually be applied
and tested. Accordingly, it is our aim to embed evolutionary theory from authors such as Lynch,
Walsh and others into biogeochemical models and carbon cycling, using a framework that is
amenable to testing and refining via experiments on the organisms that it models.

Our work demonstrates that ecological models need to represent both changes in traits (already
existing in some ecological models) and changes in the correlation between traits in order to
accurately capture phytoplankton evolution. Unlike many other evolutionary models, we do not
focus on exploring population genetic parameters like estimating population size, trait variance,
etc. We instead focused on creating a model that functions as a parallel application of theory but
in a form that informs trait-based models examining trait relationships and stabilities. This is
important because trait correlations are treated as fixed in models of marine microbes; we show
that considering evolution in trait correlations is important for understanding how phytoplankton
adapt in the face of environmental change. Similarly, while previous studies have looked at
adaptation in ODE models by ‘mutating’ parameter values such as cell size (e.g. Sauterey et al.
2015; Sauterey et al. 2017; Jiang et al. 2005; Nakov et al. 2014; Kremer and Klausmeier 2013),
maximum growth rate (Taherzadeh et al. 2017), and optimum temperature (Beckmann et al 2019),
to our knowledge, the adaptation of trait correlations have not yet been included in ODE models.
This study demonstrates that shifts in trait-correlations are fundamental for understanding the
evolved phenotype and TRACE provides a framework for evolving these relationships.
Specifically, the TRACE framework allows us to explicitly model trait adaptation using a fitness
landscape derived from easy-to-quantify trait data. This dataset can either be small (e.g. 4 traits),
such as the one we used in this study, or large ($\gg 4$ traits). Linking hypotheses of phenotype
adaptation directly to easily quantified trait values and correlations is crucial for allowing
evolutionary dynamics to be included into ecological models. We believe that this work takes a
significant step forward by providing a roadmap for including realistic adaptation dynamics in
ecosystem models. We have added in a more detailed explanation of how TRACE can be expanded
to interface with conventional ODE models of population dynamics (see entire revised Discussion
and new Supplementary Information S4 and Supplementary Fig. 8).

*Creating fitness landscapes from easy to quantify trait data*

An important advance of TRACE is that the underlying fitness landscape is constructed from
easy to quantify traits from existing high-fitness genotypes. This removes the need to rely on
constructed mutant pairs or to use theoretical fitness landscapes which can make comparisons to
real world data challenging. This is particularly important because only a small fraction of marine
microbes have been cultured (Steen et al. 2019) and those that are in culture are not genetically
tractable (with a few exceptions). As a result, traditional fitness landscapes from quantitative
genetics are difficult to apply to natural oceanographic phytoplankton communities.

Similar to dimensional reduction in quantitative genetics models, we used PCA to generate a
fitness landscape (or ‘trait scape’) that collapses the variance of N traits into two dimensions. Here
we use a simplified set of only 4 traits based on the empirical observations but the method is
expandable to as many traits as the user chooses to include. In presenting this new modeling
framework for representing trait adaptation, it was important to ensure that the results were
consistent both with empirical observations and with outcomes from prior evolutionary research.
Specifically, we are careful to highlight how the findings of TRACE are consistent with previous
evolutionary studies across fields of quantitative genetics (e.g., Anderson 1995), Fisher-based
modeling (e.g., Tenaillon 2014), and developmental bias (e.g., Uller et al. 2018).

Our work takes a huge step forward in bringing together the fields of biological oceanography
and evolutionary theory by providing a framework for understanding and constraining how
phytoplankton traits might evolve that can be directly compared against empirical data. Critically,

trait data can easily be collected for both cultured and uncultured representatives such that this
approach can have wide-reaching utility. For example, while we focus on a case study using an
experimental evolution study to generate the trait-scape, our framework can be used to generate a
trait-scape for a given environment A using data from an in situ population (assuming that the
population is well adapted to the environment they are living in). TRACE could then be used to
develop hypotheses as to how a new population from a different environment B might adapt upon
exposure to environment A.